DATA RELEASE

# Chromosome-level genome assembly and annotation of the crested gecko, *Correlophus ciliatus*, a lizard incapable of tail regeneration

Marc A. Gumangan[1], Zheyu Pan[1] and Thomas P. Lozito[1,2,*]

1 Department of Orthopaedic Surgery, Keck School of Medicine, University of Southern California, 1540 Alcazar St, Los Angeles, CA 90089, USA

2 Department of Stem Cell Biology and Regenerative Medicine, Keck School of Medicine, University of Southern California, 1425 San Pablo St, Los Angeles, CA 90033, USA

## ABSTRACT

The vast majority of gecko species are capable of tail regeneration, but singular geckos of *Correlophus*, *Uroplatus*, and *Nephrurus* genera are unable to regrow lost tails. Of these non-regenerative geckos, the crested gecko (*Correlophus ciliatus*) is distinguished by ready availability, ease of care, high productivity, and hybridization potential. These features make *C. ciliatus* particularly suited as a model for studying the genetic, molecular, and cellular mechanisms underlying loss of tail regeneration capabilities. We report a contiguous genome of *C. ciliatus* with a total size of 1.65 Gb, 152 scaffolds, L50 of 6, and N50 of 109 Mb. Repetitive content consists of 40.41% of the genome, and a total of 30,780 genes were annotated. Our assembly of the crested gecko genome provides a valuable resource for future comparative genomic studies between non-regenerative and regenerative geckos and other squamate reptiles.

**Findings:** We report genome sequencing, assembly, and annotation for the crested gecko, *Correlophus ciliatus*.

**Subjects** Genetics and Genomics, Animal Genetics, Developmental Biology

**Submitted:** 06 August 2024

\* Corresponding author. E-mail: lozito@usc.edu

Preprint submitted at https://doi.org/10.1101/2024.09.28.615630

## INTRODUCTION

The crested gecko, *Correlophus ciliatus* (NCBI:txid143539), is a lizard species endemic to New Caledonia, distinguished by eye and head projections/spines (Figure 1A, B) and the inability to regenerate amputated tails (Figure 1C, D). The chromosomes of *C. ciliatus* are typical of most New Caledonian geckos and exhibit a biarmed, acrocentric $2n = 38$ karyomorph (Figure 1E) [1, 2]. Crested geckos readily adapt to captivity; as a nocturnal, omnivorous species from a mild, tropical climate, *C. ciliatus* thrives at "room temperatures" and does not require the expensive lighting or insect diets obligatory to the maintenance of many other lizard species. Since *C. ciliatus* is able to breed nearly year-round without seasonal simulations, this species is also one of the most straightforward and productive to breed in captivity.

*C. ciliatus* is one of only fourteen described gecko species (over 1,850 total) that has lost the ability to regenerate amputated tails (Table 1). Of these non-regenerative gecko species, only *C. ciliatus* is readily available within the American and European pet hobbies.

**Figure 1.** (A) Example of a crested gecko (*Correlophus ciliatus*). (B) *C. ciliatus* head detail showing head crest spines (white arrow heads) and "eyelash" spines (black arrow heads). (C, D) Representative gross anatomy images of *C. ciliatus* and *C. sarasinorum* tails 26 days post-amputation. *C. ciliatus* tails do not regenerate like those of related gecko species, including *C. sarasinorum*. Bar = 0.25 cm. (E) *C. ciliatus* karyotype (2*n* = 38). The karyotype was prepared from *C. ciliatus* embryonic fibroblasts by the Molecular Cytogenetics Laboratory, Department of Veterinary Integrated Biosciences, Texas A&M University.

**Table 1.** Distributions and availabilities within American and European pet trades, and conservation statuses of the fourteen described geckos species lacking tail regenerative capabilities.

| Species | Distribution | Availability | Conservation status (IUCN 3.1) |
|---|---|---|---|
| *Correlophus ciliatus* | New Caledonia | ++++++ | Vulnerable |
| *Correlophus belepensis* | New Caledonia | | Critically endangered |
| *Nephrurus amyae* | Australia | +++ | Least concern |
| *Nephrurus asper* | Australia | +++ | Least concern |
| *Nephrurus sheai* | Australia | ++ | Least concern |
| *Uroplatus ebenaui* | Madagascar | +++ | Vulnerable |
| *Uroplatus fetsy* | Madagascar | | |
| *Uroplatus fiera* | Madagascar | + | |
| *Uroplatus finaritra* | Madagascar | + | |
| *Uroplatus fotsivava* | Madagascar | | |
| *Uroplatus kelirambo* | Madagascar | | |
| *Uroplatus malama* | Madagascar | | Vulnerable |
| *Uroplatus phantasticus* | Madagascar | +++ | Least concern |

Furthermore, *C. ciliatus* is the only non-regenerative lizard species capable of hybridizing with regenerative relatives, specifically *Correlophus sarasinorum*, *Mniarogekko chahoua*, and *Rhacodactylus auriculatus*. Currently, all other gecko species with sequenced genomes are capable of tail regeneration [3–7]. The goal of studying a non-regenerative gecko towards identifying gene regions involved in tail regrowth is a main driver for sequencing the *C. ciliatus* genome. With its ease of care, high productivity, and options for hybridization, the crested gecko is the ideal model lizard for studying the genetic mechanisms involved in loss of tail regeneration capabilities.

**Table 2.** General statistics of the raw sequencing reads used for the *C. ciliatus* assembly.

| Sample | Library type | Sequencing type | Platform | Number of Reads | Coverage |
|---|---|---|---|---|---|
| PacBio CCS | Long reads | Whole genome sequencing | PacBio sequel II | 13,182,509 | 62× |

## METHODS

## Sample collection, PacBio sequencing, and assembly

Gecko housing, handling, and sample collections were performed according to the guidelines of the Institutional Animal Care and Use Committee at the University of Southern California (protocol 20,992). Genomic DNA was obtained from a single whole female *C. ciliatus* embryo collected from a two-month-old egg incubated at 23 °C. The Qiagen Midi Prep Kit was used for the DNA extraction from 94 mg of the ground embryo, and approximately 100 μg of high molecular weight DNA was obtained. Genomic DNA was sequenced using the PacBio Sequel II platform (Table 2). A total of 185.8 gigabase-pairs of PacBio circular consensus sequencing (CCS) reads were used as inputs to Hifiasm v0.15.4-r347 (RRID:SCR_021069) [8] with default parameters. To estimate the genome size of *C. ciliatus*, a *k*-mer analysis was conducted on the PacBio CCS read using a range of *k* values (17, 19, 21, 23, 25, 27, 29, and 31). The estimated genome size was calculated by: (total number of *k* mers – erroneous *k* mers) divided by homozygous peak depth, following the methods of Cai *et al.* [9]. Minimum coverage was defined as the depth of the first trough in the *k*-mer frequency distribution. *K*-mers that fell under this minimum coverage were considered erroneous. Jellyfish v2.2.10 (RRID:SCR_005491) [10] was used to calculate the *k*-mer frequency using the −C parameter, and GenomeScope v1.0.0 (RRID:SCR_017014) [11] was then used to estimate heterozygosity.

BLAST v2.9.0 [12] results of the Hifiasm output assembly against the NCBI Nucleotide Database were used as inputs for blobtools v1.1.1 (RRID:SCR_017618) [13]. Scaffolds identified as possible contamination (sequences within genomic data that originate from sources other than the intended target organism, *C. ciliatus*) were removed from the assembly. Blobtools revealed contamination from Actinobacteria (two contigs, 15 Mb) (Figure 2). Finally, purge_dups v1.2.5 (RRID:SCR_021173) [14] was used to remove haplotigs and contig overlaps.

Dovetail Omni-Libraries were constructed to scaffold initial Hifiasm assemblies. For each Dovetail Omni-C library, nuclear chromatin was fixed with formaldehyde and extracted. Following digestion with DNAse I, chromatin ends were repaired and ligated to biotinylated bridge adapters followed by proximity ligation of adapter-containing ends. After proximity ligation, crosslinks were reversed, and the DNA was purified. Purified DNA was treated to remove free biotin that was not incorporated into ligated DNA fragments. Sequencing libraries were generated using NEBNext Ultra enzymes and Illumina-compatible adapters. Biotin-containing fragments were isolated using streptavidin beads before PCR enrichment of each library. Libraries were sequenced on an Illumina HiSeqX platform to produce approximately 30× sequence coverage.

The draft *de novo* assembly produced by Hifiasm and Dovetail OmniC library reads were input into HiRise (RRID:SCR_023037) [15], a software pipeline designed specifically for using proximity ligation data to scaffold genome assemblies. Dovetail OmniC library sequences were aligned to the draft assembly using BWA v0.7.17 (RRID:SCR_010910) [16]. The separations (genomic distances between pairs of reads that map within draft scaffolds)

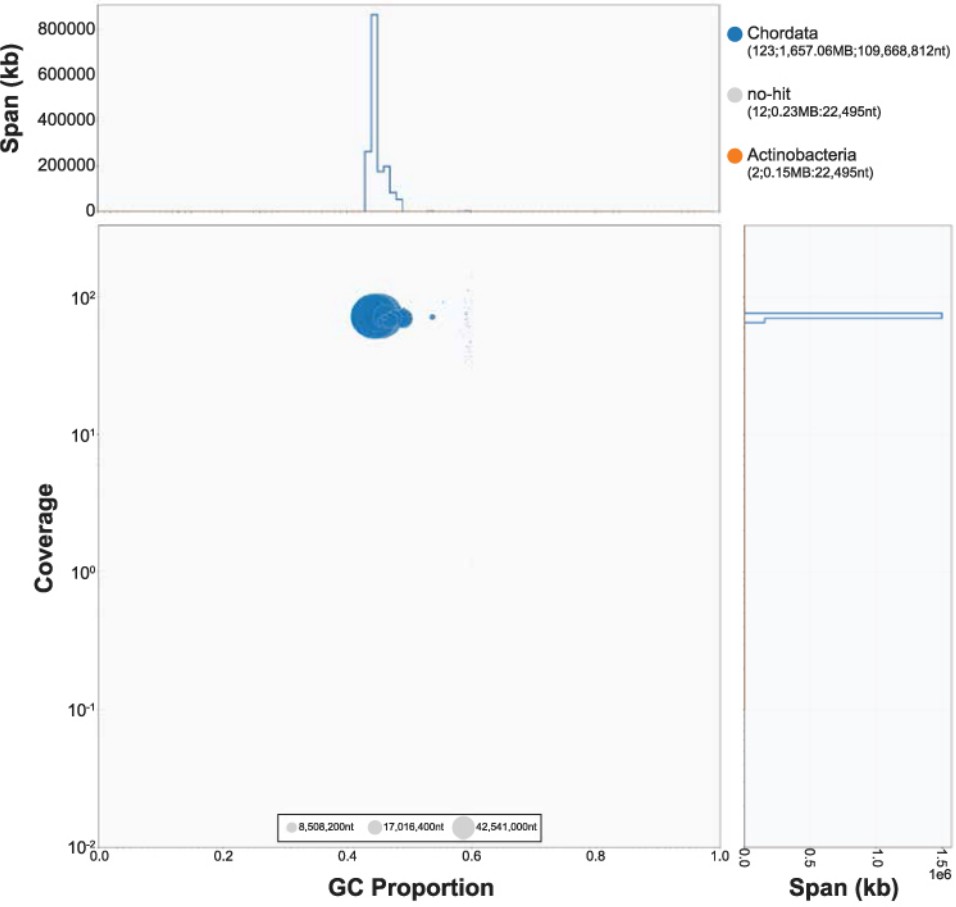

**Figure 2.** Blobtools plot showing taxonomic categories for different scaffolds (blue: Chordata, grey: 'no hits', orange: Actinobacteria), scaffold-wide coverage, and GC content. Scaffolds under Actinobacteria were removed from the final genome assembly.

of Dovetail OmniC read pairs that mapped within draft scaffolds were analyzed by HiRise to produce a likelihood model for genomic distance between read pairs. This model was used to identify and break putative misjoins (erroneous links between two contigs), to score prospective joins, and to make joins above a threshold.

### Repeat content
*De novo*-based methods were used to identify transposable elements and other repetitive elements. Repetitive content within the *C. ciliatus* genome was predicted with RepeatModeler v2.0.1 (RRID:SCR_015027) [17]. Repetitive elements were masked using RepeatMasker v4.1.0 (RRID:SCR_012954) [18].

### Gene annotation and BUSCO analysis
Coding sequences from *Anolis carolinensis*, *Gekko japonicus*, *Pogona vitticeps*, *Salvator merianae*, and *Zootoca vivipara* were used to train the initial *ab initio* model for *C. ciliatus* using the AUGUSTUS software v 2.5.5 (RRID:SCR_008417) [19]. Six rounds of prediction optimization were performed with AUGUSTUS. The same coding sequences were also used

**Table 3.** Summary of RNA-sequencing samples statistics.

| Sample ID | Library type | Sequencing type | Platform | Read length (bp) |
|---|---|---|---|---|
| geckoembryo1 | Short reads | RNA sequencing | Illumina Novaseq 6000 (PE) | 150 |
| geckoembryo2 | Short reads | RNA sequencing | Illumina Novaseq 6000 (PE) | 150 |
| geckoembryo3 | Short reads | RNA sequencing | Illumina Novaseq 6000 (PE) | 150 |
| geckoembryo4 | Short reads | RNA sequencing | Illumina Novaseq 6000 (PE) | 150 |
| geckoembryo5 | Short reads | RNA sequencing | Illumina Novaseq 6000 (PE) | 150 |
| geckoembryo6 | Short reads | RNA sequencing | Illumina Novaseq 6000 (PE) | 150 |

to train a separate *ab initio* model for *C. ciliatus* using SNAP v2006-07-28 (RRID:SCR_007936) [20]. Total RNA was extracted from a single whole, two-month-old female *C. ciliatus* embryo using the QIAGEN RNeasy Plus Kit following manufacturer protocols. Total RNA was quantified using Qubit RNA Assay and TapeStation 4200. Prior to library prep, DNase treatment was performed, followed by AMPure bead clean up and QIAGEN FastSelect HMR rRNA depletion. Library preparation was done with the NEBNext Ultra II RNA Library Prep Kit following manufacturer protocols. Then, these libraries were run on the NovaSeq6000 platform in a 2 × 150 bp configuration (Table 3). RNA-Seq reads were mapped onto the genome using the STAR alignment software v2.7 (RRID:SCR_004463) [21], and intron hints were generated with the bam2hints tools within the AUGUSTUS software. MAKER v3.01.03 (RRID:SCR_005309) [22], SNAP, and AUGUSTUS (with intron-exon boundary hints provided from bam2hints) were then used to predict gene identities in the repeat-masked reference genome. To help guide the prediction process, Swiss-Prot peptide sequences from the UniProt database [23] were downloaded and used in conjunction with the protein sequences from *A. carolinensis*, *G. japonicus*, *P. vitticeps*, *S. merianae*, *Z. vivipara* to generate peptide evidence in the MAKER pipeline. Only gene identities that were predicted by both SNAP and AUGUSTUS software were retained in the final gene sets.

## RESULTS AND DISCUSSION

The total assembly size is 1.65 Gb, with a GC content of 45% (Table 4). The estimated genome size by *k*-mer analysis is 1.52 Gb (Table 4). It is worth noting the approximately two-fold difference between the coverage of the heterozygous peak at 50X, and the homozygous peak at 100× (Figure 3). This indicates a high level of heterozygosity in the genome. The rate of heterozygosity estimated by GenomeScope is approximately 0.51% (Table 5). The contig/scaffold N50 is 109 Mb, and the largest scaffold is 169 Mbp long (Table 4). Of the total assembly, 99.54% (1,653 Mbp) was scaffolded into 19 chromosome length scaffolds (Figure 4). This number of chromosomal scaffolds is consistent with the number of haploid chromosomes observed in the *C. ciliatus* karyotype (Figure 1E).

The repetitive content consisted of 40.41% of the *C. ciliatus* genome, with a total length of 663.95 Mbp (Table 6). DNA transposons consist of 1.39%, while long interspersed nuclear element (LINE), short interspersed nuclear element (SINE), and long terminal repeat (LTR) transposons consist of 14.75%, 6.42%, and 1.08% of the genome, respectively. The *de novo* gene prediction resulted in a total of 30,780 protein-coding genes (Table 7). Of the identified genes, 20,429 (66.37%) have an Annotation Edit Distance score ≤ 0.5 (Figure 5).

## DATA VALIDATION AND QUALITY CONTROL

BUSCO v5.7.1 (RRID:SCR_015008) [24] was used to assess the quality and completeness of the *C. ciliatus* genome. BUSCO analysis was performed using the eukaryota_odb10 dataset.



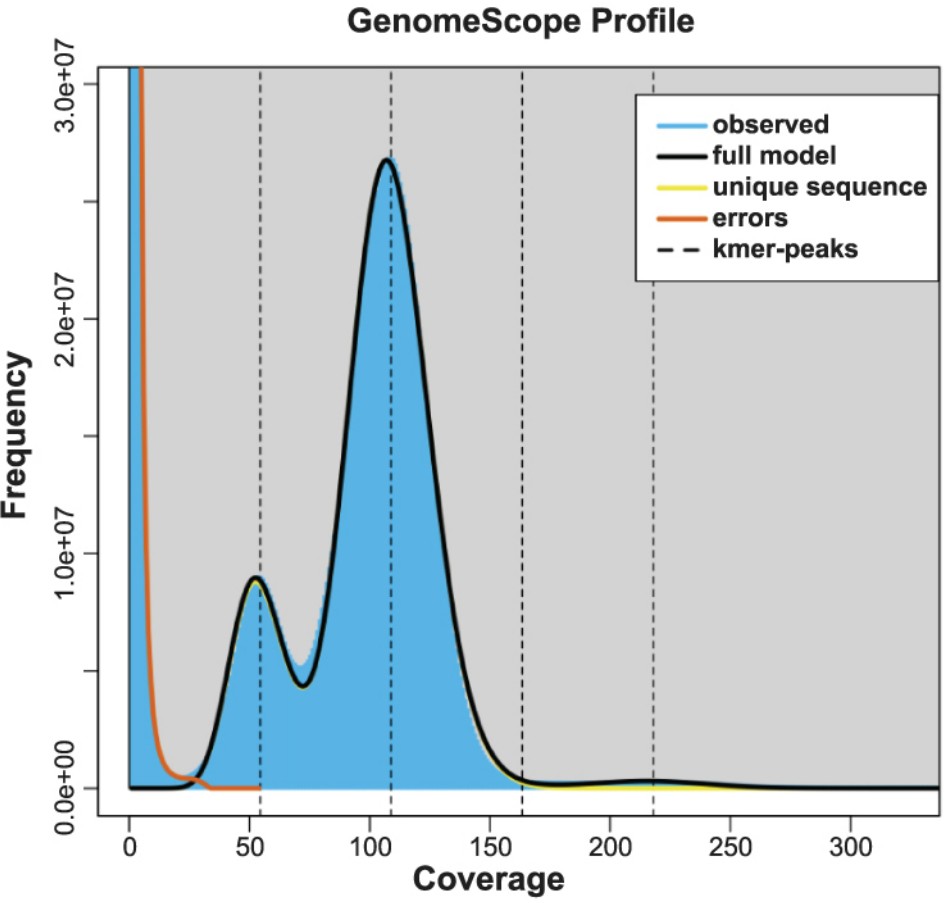

**Figure 3.** The 21-mer frequency distribution of the *C. ciliatus* genome. The first peak at 53X is the heterozygous peak, while the second peak at 107X is the homozygous peak.

**Table 4.** *C. ciliatus* genome assembly statistics.

|  | Contig | Scaffold |
|---|---|---|
| Largest scaffold length (bp) | 169,053,634 | 169,053,634 |
| N90 (bp) | 26,447,670 | 51,189,016 |
| N50 (bp) | 109,210,969 | 109,210,969 |
| L90 | 6 | 19 |
| L50 | 6 | 14 |
| Number > 1 kbp | 163 | 152 |
| Number of N's per 100 kbp | 0.00 | 0.07 |
| GC Content (%) | 45 | 45 |
| Total Size (bp) | 1,653,058,530 | 1,653,059,630 |

The *C. ciliatus* genome captured 99.6% of BUSCOs in the eukaryota_od10 dataset (Table 8), indicating the high completeness of the assembly.

## DATA AVAILABILITY

Supporting datasets, including annotation, are available at GigaDB [25]. Raw sequencing reads and OmniC Library reads have been deposited in the Sequence Read Archive

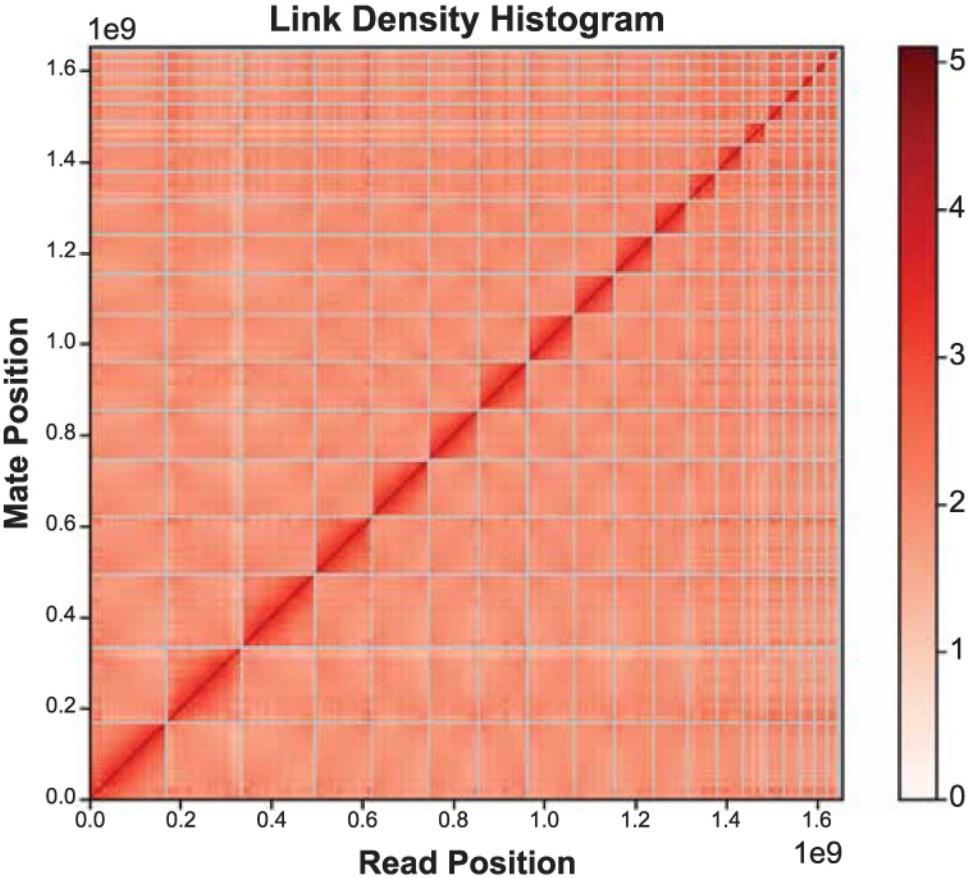

**Figure 4.** Hi-C contact plot of 19 chromosomal scaffolds, along with unassigned scaffolds.

**Table 5.** Estimation of genome size and heterozygosity of the *C. ciliatus* assembly.

| k | Total number of k-mers | Minimum coverage (X) | Number of erroneous k-mers | Homozygous peak | Heterozygous peak | Estimated genome size (Gb) | Estimated heterozygosity (%) |
|---|---|---|---|---|---|---|---|
| 17 | 160,830,997,409 | 23 | 1,309,311,311 | 110 | 55 | 1.45 | 0.46 |
| 19 | 163,207,828,191 | 21 | 2,876,102,102 | 108 | 53 | 1.48 | 0.51 |
| 21 | 164,978,566,184 | 21 | 3,624,290,072 | 107 | 53 | 1.51 | 0.52 |
| 23 | 166,415,784,027 | 21 | 4,147,151,401 | 107 | 53 | 1.52 | 0.52 |
| 25 | 167,625,136,836 | 21 | 4,629,779,608 | 107 | 52 | 1.52 | 0.52 |
| 27 | 168,655,291,501 | 21 | 5,100,636,735 | 106 | 52 | 1.54 | 0.51 |
| 29 | 169,546,294,955 | 21 | 5,566,480,734 | 106 | 52 | 1.55 | 0.50 |
| 31 | 170,331,212,990 | 21 | 6,029,874,685 | 106 | 52 | 1.55 | 0.50 |

database under Bioproject ID PRJNA1091669. RNA-Seq reads have been deposited under BioProject ID PRJNA1128839. This Whole Genome Shotgun project has been deposited at DDBJ/ENA/GenBank under the accession JBBPXQ000000000, and the Biosample accession number is SAMN40604022. The version described in this paper is version JBBPXQ010000000.

## ABBREVIATIONS
AED, annotation edit distance; CCS, circular consensus sequencing; LINE, long interspersed nuclear element; LTR, long terminal repeat; SINE, short interspersed nuclear element.

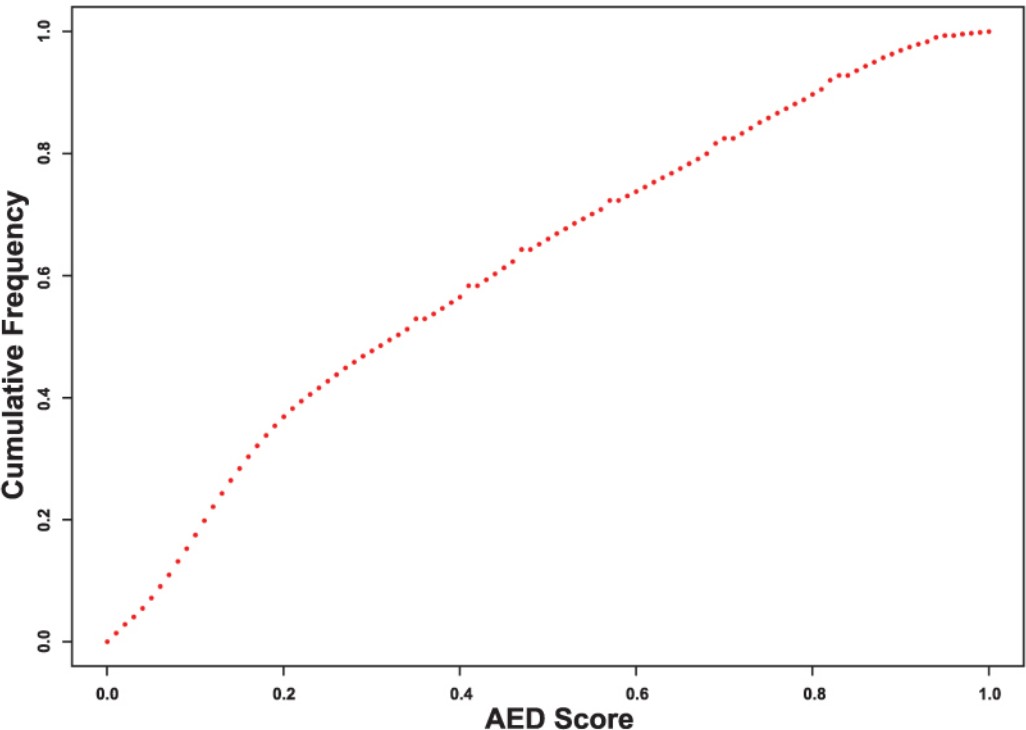

**Figure 5.** The frequency scores of the annotation edit distance (AED) for the *Correlophus ciliatus* assembly. The AED is a general measure of how well the predicted gene is supported by external evidence (UniProt protein and mRNA sequences). The AED score ranges from 0 to 1, and a lower score represents more evidence support for the gene. The AED is calculated for every gene. The AED cumulative frequency graph provides an overview of the quality of the gene annotation.

**Table 6.** Summary of repetitive content and transposable elements of the *C. ciliatus* genome assembly.

| Type | Number of elements | Length (bp) | % of genome |
|---|---|---|---|
| DNA | 173,526 | 23,044,151 | 1.39 |
| LINE | 789,302 | 243,834,212 | 14.75 |
| SINE | 652,803 | 106,143,112 | 6.42 |
| LTR | 58,109 | 17,890,933 | 1.08 |
| Other | 889 | 381,767 | 0.02 |
| Unknown | 1,902,327 | 272,947,662 | 16.51 |
| Small RNA | 2,588 | 3,815,594 | 0.23 |
| Satellites | 1,039 | 952,457 | 0.06 |
| Simple repeats | 292,097 | 14,087,986 | 0.85 |
| Low complexity | 25,303 | 1,434,776 | 0.09 |
| Total | 3,897,983 | 684,532,650 | 40.41 |

**Table 7.** *De novo* gene prediction metrics.

| | |
|---|---|
| Total number of genes | 30,780 |
| Total coding regions (bp) | 39,760,070 |
| Average length of genes (bp) | 1,291.75 |
| Number of single-exon genes | 1,461 |

**Table 8.** BUSCO analysis summary of the *C. ciliatus* genome.

| BUSCO Category | Value | Percent (%) |
|---|---|---|
| Complete BUSCOs | 254 | 99.6 |
| Complete and single-copy BUSCOs | 249 | 97.6 |
| Complete and duplicated BUSCOs | 5 | 2 |
| Fragmented BUSCOs | 1 | 0.4 |

## DECLARATIONS

### Ethics approval and consent to participate

The authors declare that ethical approval was not required for this type of research.

### Competing interests

The authors declare that they have no competing interests.

### Authors' contributions

TL conceived and supervised the project and provided the crested gecko samples. MG analyzed the genome assembly and performed the repeat annotation and the BUSCO analysis. MG drafted the manuscript. TL revised the manuscript. ZP maintained animal colonies. All authors read and approved the final manuscript.

### Funding

We would like to acknowledge funding from NIH R01GM115444 and support from the CIRM COMPASS Award (EDUC5-13853). No additional external funding was used.

### Acknowledgements

Special thanks to Dr. Andrew McMahon, the Department of Stem Cell Biology and Regenerative Medicine, and the Eli and Edythe Broad Center for Regenerative Medicine and Stem Cell Research at the University of Southern California for genome sequencing support.

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
