## [Editor Report]

Editor’s AssessmentThe crested gecko (Correlophus ciliatus), is a lizard species endemic to New Caledonia, and a potentially interesting model organism due to its unusual (for a gecko) inability to regenerate amputated tails. With that in mind here is presented a new reference genome for the species, assembled using PacBio Sequel II platform and Dovetail Omni-C libraries. Producing a genome with a total size of 1.65 Gb, 152 scaffolds, a L50 of 6, and N50 of 109 Mb. Peer review making sure more detail was added on data acquisition and processing to enhance reproducibility. In the end producing potentially useful data for studying the genetic mechanisms involved in loss of tail regeneration.Editor’s AssessmentThe crested gecko (Correlophus ciliatus), is a lizard species endemic to New Caledonia, and a potentially interesting model organism due to its unusual (for a gecko) inability to regenerate amputated tails. With that in mind here is presented a new reference genome for the species, assembled using PacBio Sequel II platform and Dovetail Omni-C libraries. Producing a genome with a total size of 1.65 Gb, 152 scaffolds, a L50 of 6, and N50 of 109 Mb. Peer review making sure more detail was added on data acquisition and processing to enhance reproducibility. In the end producing potentially useful data for studying the genetic mechanisms involved in loss of tail regeneration.

---

## [Reviewer Report]

Reviewer name and names of any other individual's who aided in reviewer Anthony Geneva and Cleo FalveyDo you understand and agree to our policy of having open and named reviews, and having your review included with the published papers. (If no, please inform the editor that you cannot review this manuscript.)YesIs the language of sufficient quality?YesPlease add additional comments on language quality to clarify if needed
Are all data available and do they match the descriptions in the paper? YesAdditional CommentsAre the data and metadata consistent with relevant minimum information or reporting standards? See GigaDB checklists for examples <a href="http://gigadb.org/site/guide" target="_blank">http://gigadb.org/site/guide</a>YesAdditional CommentsIs the data acquisition clear, complete and methodologically sound?YesAdditional CommentsIs there sufficient detail in the methods and data-processing steps to allow reproduction?YesAdditional CommentsIs there sufficient data validation and statistical analyses of data quality? YesAdditional CommentsIs the validation suitable for this type of data?YesAdditional CommentsIs there sufficient information for others to reuse this dataset or integrate it with other data?YesAdditional CommentsAny Additional Overall Comments to the AuthorIn their revised manuscript Gumangan and colleagues have addressed each of the comments we made on the original manuscript via substantial revisions. We appreciate the improvements the authors have made but feel there are a few remaining issues that require attention, detailed below. Those issues notwithstanding, this new assembly and annotation represent a very nice contribution to the field and will certainly be widely used. Specific comments: Pages 2 and 6: Each time L50 or L90 statistics are reported they are listed with the units “bp”. These values are counts of scaffolds are are typically simply reported as integers without units. Page 3. “Furthermore, C. ciliatus is the only non-regenerative lizard species capable of hybridizing with regenerative relatives, specifically C. sarasinorum, Mniarogekko chahoua, and Rhacodactylus auriculatus.” This statement is very interesting but requires a reference or at least attribution of some kind (eg - personal observation by one of the co-authors). Page 3: “Genomic DNA was sequenced using the Illumina Novaseq 6000 platform. 185.8 gigabase-pairs of PacBio CCS reads were used as inputs to Hifiasm v0.15.4-r347 [8] with default parameters.” The sequencer listed here for generating long reads seems to be an error and should be some PacBio platform (Sequel, Sequel IIe, etc). Page 6: “The contig/scaffold N50 is 109 Mb, and the largest scaffold had a length 1169 Mbp (Table 1)”. 1169 should be 169.RecommendationMinor Revision

---

## [Reviewer Report]

Upload additional filesDRR-202408-01-R01/stage_files/DRR-202408-01/Review MS/review.docxReviewer name and names of any other individual's who aided in reviewer Zexian ZhuDo you understand and agree to our policy of having open and named reviews, and having your review included with the published papers. (If no, please inform the editor that you cannot review this manuscript.)YesIs the language of sufficient quality?YesPlease add additional comments on language quality to clarify if needed
Are all data available and do they match the descriptions in the paper? YesAdditional CommentsAre the data and metadata consistent with relevant minimum information or reporting standards? See GigaDB checklists for examples <a href="http://gigadb.org/site/guide" target="_blank">http://gigadb.org/site/guide</a>NoAdditional CommentsIs the data acquisition clear, complete and methodologically sound?YesAdditional CommentsIs there sufficient detail in the methods and data-processing steps to allow reproduction?YesAdditional CommentsIs there sufficient data validation and statistical analyses of data quality? YesAdditional CommentsIs the validation suitable for this type of data?YesAdditional CommentsIs there sufficient information for others to reuse this dataset or integrate it with other data?YesAdditional CommentsSee the attachmentAny Additional Overall Comments to the AuthorRecommendationMinor Revision

---

## [Reviewer Report]

Reviewer name and names of any other individual's who aided in reviewer Chaochao YanDo you understand and agree to our policy of having open and named reviews, and having your review included with the published papers. (If no, please inform the editor that you cannot review this manuscript.)YesIs the language of sufficient quality?YesPlease add additional comments on language quality to clarify if needed
Are all data available and do they match the descriptions in the paper? NoAdditional CommentsIn the section "Availability of supporting data," it is stated that "supporting datasets, including annotation, are available at GigaDB." However, I was unable to locate these datasets during my search. Could you please provide a direct link or the accession number to access these resources?Are the data and metadata consistent with relevant minimum information or reporting standards? See GigaDB checklists for examples <a href="http://gigadb.org/site/guide" target="_blank">http://gigadb.org/site/guide</a>YesAdditional CommentsIs the data acquisition clear, complete and methodologically sound?NoAdditional CommentsThe manuscript currently lacks detailed information regarding the samples and data used to assemble and annotate the reference genome. For instance, it does not specify how many samples or libraries were used for RNA-Seq or whole-genome sequencing. I suggest including a table that provides comprehensive information on the samples and sequencing data. Additionally, while the manuscript mentions that "Genomic DNA was sequenced using the Illumina Novaseq 6000 platform," the corresponding Illumina data are not described. I am unclear about how the PacBio CCS reads were produced. Could you please provide more details or clarify the methodology used to generate these reads?Is there sufficient detail in the methods and data-processing steps to allow reproduction?NoAdditional CommentsSome methods described in the manuscript lack sufficient detail, particularly for tools such as BLAST, BlobTools, HiRise, and BWA. To ensure reproducibility, I recommend providing the specific parameters used for these analyses.Is there sufficient data validation and statistical analyses of data quality? YesAdditional CommentsIs the validation suitable for this type of data?YesAdditional CommentsIs there sufficient information for others to reuse this dataset or integrate it with other data?YesAdditional CommentsAny Additional Overall Comments to the AuthorRecommendationMinor Revision